# Driver Behavior When Overtaking Cyclists Riding in Different Group Configurations on Two-Lane Rural Roads

**DOI:** 10.3390/ijerph182312797

**Published:** 2021-12-04

**Authors:** Ana María Pérez-Zuriaga, Sara Moll, Griselda López, Alfredo García

**Affiliations:** Highway Engineering Research Group, Universitat Politècnica de València, Camino de Vera sn, 46022 Valencia, Spain; samolmon@upv.es (S.M.); grilomal@tra.upv.es (G.L.); agarciag@tra.upv.es (A.G.)

**Keywords:** overtaking cyclists, two-lane rural road, safety, geometric design, instrumented bicycle, cyclist group

## Abstract

The presence of cyclists on Spanish rural roads is ever increasing and currently frequent, and thus becoming a serious safety concern. In rural environments, the risk of a crash is higher than in rural areas. The main cause is the higher speed of motor vehicles during overtaking manoeuvres. This manoeuvre is especially challenging when cyclists ride in groups as they may change size, length, shape, and speed along their route. These variables and those related to road cross-section can influence driver behaviour when overtaking a group of cyclists. To study this, instrumented bicycles were used to ride along five road segments with different geometric and traffic characteristics. Cyclists rode individually and in groups. Overtaking was evaluated by analysing the lateral distance, the speed, and other characteristics of the manoeuvre. Wider roads presented higher lateral clearances and overtaking speeds. Narrower roads had a high opposing lane invasion but a high level of compliance with the minimum lateral clearance. A higher clearance and lower speed of overtaking vehicles was registered when cyclists rode in line. Compliance with the 1.5 m clearance depended on the group configuration, being higher when cyclists rode in line. However, overtaking cyclists riding two abreast presented more accelerative manoeuvres, especially on narrow roads.

## 1. Introduction

Cycling is increasingly popular in Europe, and particularly in Spain, both in rural and urban areas. However, whereas in urban areas it is related with commuting to work or school, cycling in rural areas is mostly related to sport and leisure activities. Moreover, due to the lack of segregated bike lane on most two-lane rural roads and the features of the existing ones, large cyclist groups usually ride sharing the road with motorized vehicles. This situation increases the risk of accident.

In 2018, in the EU, 2160 cyclist deaths were recorded in traffic collisions, including single bicycle collisions with no other vehicle involved or falls after an interaction with another road user that did not actually end in a physical contact, although 83% of all cyclist deaths are a consequence of an impact with a motor vehicle [1]. The number of cyclist deaths has decreased by only 0.4% on average each year in the EU over the period from 2010 to 2018, compared with a 3.1% annual reduction in motorized road user deaths over the same period. Regarding the distribution of fatalities within urban areas and in rural areas, there are large differences between countries.

In Spain, in that year the number of accidents with victims involving cyclists was 8095:2426 (30%) in rural areas and 5669 (70%) in urban areas. However, crash reports show that severity is much higher on rural roads, where 54 (72%) fatal accidents occurred inside rural areas, whereas 21 (28%) occurred inside urban areas [2]. The main reason for this difference is the higher speed of motor vehicles, specifically the higher relative speed between motor vehicles and bicycles during overtaking maneuvers. The risk increases when the passing vehicle is a heavy goods vehicle [3].

Thus, Spanish traffic regulations [4] indicate that cyclists must ride as close as possible to the outer edge of the road and motor vehicles overtaking cyclists must keep a minimum lateral distance of 1.5 m.

Similarly to Spain, most countries regulations establish a minimum lateral separation which does not depend on the speed or on vehicle type. This minimum lateral separation varies from 1 m, as in some U.S. States, to 1.5 m, as in Spain. In other regions the established minimum lateral distance depends on motor vehicle speed. For instance, in Queensland (Australia), the Department of Transport [5] recommends 1 m if the motor vehicle travels under 60 km/h and 1.5 over 60 km/h. In order to comply with this law, drivers overtaking cyclists are exempt from the general prohibitions on driving over center lines on two-lane rural roads.

It is usually not clear whether the gap is between the body of the car and the bicycle (lateral spacing), or considering the handlebar of the bike and the rear-view mirror of the vehicle (lateral clearance). Llorca et al. [6] analyzed the compliance of the Spanish 1.5 m regulation, concluding in 9% of noncompliance if lateral spacing is considered and in 36% of noncompliance if the considered gap is the lateral clearance. Even though the lateral passing distance is considered as the lateral spacing (aerodynamic effect) or the lateral clearance (collision risk), this variable is a key determinant of cyclist safety.

The lateral passing distance has been studied based on data collected through five main data collection methodologies: external video recording, instrumented bicycles, driving simulator, naturalistic driving studies [7], and test-track experiments [8]. In most of these studies, speed of the motor vehicles was also considered since it contributes to both aerodynamic forces which can compromise cyclists’ stability and crash severity [6].

Video recording with high-definition cameras has been used to assess the lateral placement of motor vehicle when overtaking bicycles, concluding that drivers were more likely to contact or cross over the centerline the nearer the cyclists were to the travel lane, and that the presence of opposing traffic reduce the likelihood of contact with the centerline [9]. It has also been used to examine the influencing motorists’ compliance with legislated passing distance rule. Higher non-compliance rates were found on higher speed roads, at curves, on roads with bike lanes, and roads without footpaths, but greater compliance rates were observed on wider traffic lanes [10].

Video recording is unobtrusive and allows data collection without influencing road users’ behavior. However, the location of the cameras may impact the viewing angle and limit the precision of the lateral positioning of the vehicles. Moreover, video cameras can only be located in specified sites, thus limiting the generalizability of the findings. Therefore, several studies have used instrumented bicycles to analyze overtaking maneuvers.

Studies based on this data collection concluded that the further the rider was from the edge of the road, the closer vehicles passed [11]. It has also been used to study the influence of roadway geometric elements on driver behavior when overtaking bicycles on rural roads. Their results showed that lateral clearance distance increases with road speed limit, when marking centerline exists, with the shoulder width and the bicycle speed, and decreases in downhill road grade sections and in the presence of opposing vehicles [12,13]. A study carried on in Spain concluded that the passing lateral clearance increased with road width [6,14]. Moreover, it was higher on left curve and lower on right curve, compared with the tangent elements. Furthermore, variables affecting the risk perception of the cyclist were analyzed [6]. They found that lateral clearance was not the only factor that influenced rider’s risk perception. A combined factor of lateral clearance, vehicle type and vehicle speed had a more significant correlation with it.

The influence of the presence of a cycle lane on the lateral clearance was also analyzed. The results suggested that at higher speeds and in absence of a cycle lane, motor vehicles provide more passing distance than in roads where the cycle lane is insufficiently wide [15]. However, other studies concluded also that the passing distance was reduced when the cyclists were riding in a marked on-road bicycle lane, but they concluded that it did not vary between speed zones [16].

Instrumented electric bicycle has been used to recording data to identify the overtaking phases: approaching, steering away, passing, and returning [17]. They also quantified the corresponding driver comfort zones measured as the distance between the bicycle and the vehicle (instead of only for the passing phase, as with previous studies). The presence of an oncoming vehicle was the factor which most influenced the maneuver, whereas neither vehicle speed, lane width, shoulder width, nor posted speed limit significantly affected the driver comfort zone or the overtaking dynamics. Afterwards, two different time-to-collision (TTC) measures were added to the quantification of the drivers’ comfort zone boundaries (CZB) [18]. They concluded that the higher the car speed the larger the CZBs while approaching and passing, but the presence of an oncoming vehicle significantly decreased the CZB during passing.

In order to increase the sample size, other studies [19] have based their study of the bicycle overtaking maneuver on events extracted from the naturalistic driving study Safety Pilot Model Deployment. The variable used as a surrogate safety measure was the vehicle lateral placement at the time of passing bicyclists and studied the influence of several factors, such as: lane marking type, the presence of a bike lane or paved shoulder, the presence of traffic, lane width, and driver distraction.

These naturalistic studies cannot control several variables that might affect the results. More control can be achieved in simulator studies. These studies have focused on the effects of cross-section configuration on drivers’ behavior during the interaction with a cyclist without oncoming traffic, the differences when cyclists overtaking is carried out on tangent, on left curve and on right curve [20] and on how the oncoming traffic affects the overtaking strategy [21,22]. The main limitations of this simulator studies are that drivers’ behavior may differ from on road driving and that the estimation of the distance to the oncoming traffic and the corresponding time to collision may also differ from reality. In addition, these studies are based on a small sample size.

Most of these studies only studied drivers’ behavior when overtaking a bicycle riding alone. The phenomenon may be different when overtaking a group. In fact, studies concluded that a driver is more likely to cross over the centerline into the opposing lane when overtaking bicyclists riding in a group of two or more cyclists [9]. In Spain, a study compared different group configurations riding on a two-lane rural road segment: a single bicycle, two bicycles in parallel, two in line, and different groups of three cyclists. They equipped the bicycles with cameras and laser rangefinders and also registered the subjective risk perception at every overtaking event. They found that when cyclists rode two abreast, drivers overtaking leave lower lateral clearance, and that the overtaking vehicle speed is higher when individual cyclist is passed [23]. López et al. [24] increased the number of cyclists in the group to 10 cyclists, riding in different configurations. They analyzed the objective and the subjective risk of overtaking maneuvers to cyclists’ groups, using as surrogate measures the speed and the lateral distance. Results showed that the risk is higher when cyclists ride in parallel lines and when the overtaking maneuver is flying.

Regardless of the data collection methodology used, the main objective of most research is the characterization of the interaction between a motor vehicle and a bicycle during overtaking maneuver. However, only a few studies have assessed the overtaking of a cyclists’ peloton. The present study presents an approach to drivers’ behavior when overtaking a peloton. The main objective is to analyze the effect of several variables related to infrastructure on drivers’ behavior when overtaking a single cyclist, a medium group of cyclists, and a peloton.

## 2. Materials and Methods

This study aims to analyze the behavior of drivers when overtaking cyclists riding individually and in medium and large groups. For this purpose, instrumented bicycles were used to collect data on lateral clearance and speed of the overtaking vehicle as well as other variables during overtaking maneuvers. Sport cyclists have performed several rides individually and in medium and large groups with different configurations. The study was carried out on five segments of two-lane rural roads with different traffic and geometric characteristics.

### 2.1. Road Sections

Research team carried out previous research focused on estimating cyclist demand of two-lane rural roads. One of the results was the location of several two-lane rural road segments in Valencian Region (Spain) with high cyclist demand. Five of these road segments were chosen to collect data on overtaking maneuvers (Figure 1).

This choice is related to the different characteristics that infrastructure and traffic have (see Table 1): different lane and shoulder width, colored or uncolored shoulder, different AADT (Average Annual Daily Traffic).

As shown in Table 1, the road with highest traffic is RS1 (its AADT is more than twice that of the other roads).

The lane width has a difference of 0.5 m among all the roads analyzed, with RS1 being the road with the narrowest lanes. However, analyzing the half carriageway width (lane and shoulder) the narrowest is RS2. According to road shoulder, the most different road segments are: RS2 that has not shoulder; RS3 that has a red colored shoulder; and RS4 that has a 1 m shoulder with lateral barriers (Figure 1).

Speed limit also differs between the road segments. The road with lower speed (RS1) also has a speed control with radar. The road with highest speed limit is the RS5, during the period in which data collection was performed, the speed was limited to 100 km/h; later, the DGT made a revision of the speed limits on rural roads in Spain, and its current limit is 90 km/h.

### 2.2. Cyclist Group Configurations

During the experimental design phase, the different cyclist configurations in which the study sections would be ridden were defined. The defined configurations are shown in Figure 2. They range from a single cyclist to group of cyclists of medium size (4 cyclists) and large size (10 cyclists). In addition, the cyclists rode in the different configurations allowed by the regulations in Spain: in-line and two abreast. The election of these group configurations is based on the results of a study carried out in the initial stages of the research projects, where it was found that these were the most usual configurations.

### 2.3. Instrumented Bicycles

Experiment was performed using partially and/or fully instrumented bicycles, depending on the cyclist’s configuration. Fully instrumented bicycles are shown in Figure 3. They were equipped with a Laser Technology Inc. (Centennial, Colorado, CO, USA) T200/T100 laser system. It consisted of a couple of laser rangefinders, perpendicular to bicycle axis, one at the front part of the bicycle and the other at the rear. According to the device specifications, the T100 uses infrared laser light to measure distance, with a pulse repetition frequency up to 25 kHz. This invisible light is emitted from the transmit lens of the sensor, reflected off the vehicle and returns to the receive lens of the sensor. The exact distance is then calculated by the system by comparing the return time to the speed-of-light constant. Hence, the distance from the bicycle to the overtaking vehicle body is measured. The two sensors system measures the speed when a vehicle passes the sensors, going from the T100 to the T200, after computing the time interval between the measurements of the two rangefinders, with a speed accuracy of 2%. Considering that the bicycle is moving, the sensors system measures the relative speed between vehicle and bicycle.

Bicycles also have two high-definition video cameras (Garmin VIRB Élite) recording information on the environment. The forward-facing video camera allowed registering the presence of opposing vehicles, and the rear-facing video camera monitored the approach of the overtaking vehicle. The video cameras contain a 1 Hz GPS to obtain the position of the bicycles and bicycle speed. Overtaking speed is obtained considering the speed data measured by the T200/T100 laser system and the bicycle speed obtained from camera GPS data.

Fully instrumented bicycles (shaded in grey in Figure 2) were used by individual riders and in the rear and front positions for groups of riders. The remaining bicycles were partially instrumented with the two high-definition video cameras.

### 2.4. Data Collection and Reduction

Data collection days and time periods were selected based on previous observations of cyclist traffic on each road section. A total of 24 data collection sessions were conducted. Data collection for groups of 10 cyclists was conducted on weekends, whereas for individual cyclists and groups of 4 cyclists, weekdays were chosen. Data collection time periods were those when peak cycle demand was observed in previous research [25]. Two data collection sessions were conducted on each road segment, and finally, an additional data collection was conducted to complete sufficient data on overtaking maneuvers on each road section and for each cyclist group configuration.

Data were collected and recorded by video cameras and laser devices placed on the bicycles. Data reduction was performed using specific video review software and synchronization of the laser device data. In this way, a complete data set including several variables for each overtaking maneuver was obtained.

### 2.5. Study Variables

From the GPS of the video cameras were obtained the bicycle speed (Bs). Laser device registered lateral distance (Ld) and relative speed (S), then the lateral clearance and the overtaking speed were calculated as follows:Lateral clearance (m) = Ld − half handlebar width (0.15 m) − vehicle side mirror width (0.12 m).Overtaking vehicle speed (km/h) = S + B_s._

In four and ten cyclists’ groups, where two bicycles were fully instrumented, two overtaking speed data and two lateral clearance data were recorded. The highest overtaking speed and the lowest lateral clearance for each single maneuver were considered for the analysis.

From the review of the videos other variables related to environment were obtained:Overtaking vehicle type: passenger car (PC), motorcycle (MC) or heavy vehicle (HV).Half carriageway width (m): lane + shoulder width.Speed limit of each road segment (km/h).Overtaking maneuver strategy: Flying strategy (driver overtakes cyclists while keeping their speed relatively constant); accelerative strategy (driver slows down and follow the cyclists for some time before passing); piggy backing strategy (drivers who follow a lead driver, so that two or more cars in a row overtake the cyclists).Centre line: solid line or dashed line.Horizontal alignment: tangent, right curve or left curve.Invasion of opposing lane: no invasion, partial invasion or total invasion.

## 3. Results

This section includes the main results obtained in this study. First of all, the total overtaking events recorded is shown. Anomalous data due to malfunctioning of the devices, overtaking maneuvers in which cyclists did not respect the group configuration or performed at intersections were discarded. Therefore, a total of 9580 interactions between cyclists and drivers, considering both overtaking and crossing events, were registered. From these, 1580 were overtaking events (Table 2).

Although data collection attempted to obtain a sufficient number of overtaking maneuvers to conduct the study considering all road segments and all cyclist group configurations, Table 1 shows that a higher number of overtaking maneuvers were recorded on roads with high AADT. Regarding cyclist group configurations, more overtaking maneuvers of an individual cyclist were recorded, with the group of 10 cyclists being the one with the lowest number of overtaking maneuvers. These results indicate the higher difficulty in overtaking large groups of cyclists, as well as overtaking on roads with a high level of traffic.

### 3.1. Overtaking Vehicle Type

The type of overtaking vehicle was recorded for each overtaking maneuver by classifying them into several vehicle types. For this study, vehicles were regrouped into three general types: passenger cars (PC), heavy vehicles (HV), and motorcycles (MC). During data collection, 70 overtaking maneuvers were performed by heavy vehicles, 56 by motorcycles, and 1442 by passenger cars. Only 1359 and 1350 maneuvers out of the 1442 maneuvers performed by passenger car were valid for lateral clearance study and speed analysis, respectively, due to failures in data collection devices (see Table 3).

To determine the differences between types of vehicles on overtaking cyclists, the lateral distance and the speed of the overtaking vehicle for each overtaking maneuver were analyzed.

Table 3 shows that HV overtakes keeping a lower lateral clearance, whereas the higher mean value of lateral clearance was registered for MC. Regarding overtaking vehicle speed, MC presented the higher speed, whereas PC and HV presented a similar speed.

An ANOVA test and Fisher’s LSD method were performed to determine if significant statistical differences exist between different vehicle types. Results for clearance (*F* = 4.01, *p*-value = 0.018) and speed (*F* = 7.14, *p*-value = 0.001) showed significant statistical differences between motorcycles and the other types of vehicles with a 95% confidence level indicating higher lateral clearances and higher overtaking speeds for MC. Although passenger cars and heavy vehicles did not show significant statistical differences in speed and clearance, their mean values were different.

Considering these results and the lower number of overtaking maneuvers registered of HV and MC, the study was performed considering only overtaking maneuvers performed by PC.

### 3.2. Influence of Road Section Characteristics on Lateral Clearance

The lateral clearance distance left by a driver when overtaking a cyclist or cyclists’ group could be assumed as one of the more important variables to consider from the road safety point of view. This variable has an important influence on collision occurrence.

To study this variable, a descriptive analysis was performed to show up how it varies depending on other variables, such as the road segment and the group configuration. When considering the road segment, variables related to infrastructure and traffic are also analyzed. Figure 4 shows the box-and-whisker plots for this variable, including the 1.5 m minimum distance, according to Spanish regulations.

RS1 and RS3 presented lower lateral clearance than other road segments, which correspond to the road segment with higher AADT (RS1) and the road segment with the red-colored shoulder (RS3). The road segment with higher lateral clearance was RS5, which is the one with higher width, considering both shoulder and lane.

An additional statistical analysis was performed using Fisher’s LSD intervals. This test allows determining of significant differences between individual groups. Moreover, the LSD intervals show the intervals for which a statistically significant difference between the means can be assumed. Figure 5 shows the results of this study applied to the lateral clearance. Analyzing the variation on lateral clearance, it can be assumed that:RS1: on this road, when 1 cyclist rode individually, the lowest lateral clearance was recorded. In the groups of medium and large cyclists, a greater lateral clearance was recorded when riding in line. There are significant differences only in the case of 1 individual cyclist and 4 cyclists in line.RS2: in the case of 4 cyclists riding two abreast, there were significant differences in lateral clearance compared with the configurations of the other groups of cyclists. A higher clearance was obtained when the groups of cyclists rode in line.RS3: statistically significant differences were only obtained for a medium group (4 cyclists) riding in line and two abreast. On this road, when the groups of cyclists rode in line, greater clearances were observed than when two abreast.RS4: no statistically significant differences were observed between the different configurations. On this road, unlike the others, when the groups rode in line, the lateral clearances were lower than when they rode two abreast.RS5: no statistically significant differences were observed between the different configurations. On this road, practically the same clearances were recorded for all cyclist configurations.

In general, the mean values of the lateral distances were higher when the groups of cyclists were riding in line, especially on the narrow RS2 and the colored shoulder RS3 roads. On RS4, which is a narrow road with lateral barriers, the effect was the opposite, the lateral clearance was higher when overtaking groups riding two abreast. On the other hand, RS5, with a wider cross section and a higher speed limit, did not show differences in lateral clearances between the different cyclist grouping configurations offering the higher mean values of lateral clearance.

Additionally, a multivariable regression modeling process of the lateral clearance for each cyclists’ group configuration was carried out, considering as independent variables: the different road segments (e.g., RS1 value is 1 when the passing maneuver occurs in road segment 1 and 0 otherwise), being the RS5 the referent; the centerline type (e.g., Solid line value is 1 when the centerline is solid and 0 when it is dashed); passing maneuver type (e.g., Accelerative value is 1 when the overtaking strategy is accelerative and 0 otherwise). RS5 has been chosen as referent since this road segment presents lower restrictions: wider lane and shoulder, higher speed limit and lower AADT.

Moreover, the effect of the location of the passing maneuver (tangent, right curve and left curve) was studied, but it was not significant for all the cyclists’ group configurations.

The results of the model (Table 4) shows that the effect of the overtaking strategy is only significant when a cyclist is riding alone, increasing the lateral clearance when the maneuver is accelerative. The type of center line is only significant when the group configuration is 10 TA. In that case, the lateral clearance decreases when the center line is solid.

According to road segment, when the cyclists’ group is large (4 L, 10 L and 10 TA), the results are quite similar among road segments. In RS3, which has the red-colored shoulder, the lateral clearance decreases when a cyclist is riding singly and when the cyclists ride two abreast.

### 3.3. Influence of Road Section Characteristics on Overtaking Vehicle Speed

A similar analysis was developed for the overtaking vehicle speed, which is the most important variable from the point of view of the severity of crashes.

Figure 6 shows the box-and-whisker plots for this variable, including the speed limits of each road segment.

Road segments with lower speed limit presented an overtaking speed next to the speed limit. However, road segments with higher speed limits presented an overtaking speed lower than the speed limit. From these results, it could be assumed that drivers overtaking cyclists at not excessively high speeds. This behavior may be due to the vulnerability of cyclists, especially on rural roads.

This descriptive analysis has been completed with a statistical analysis using Fisher’s LSD intervals. Figure 7 shows the results of this study applied to the vehicle overtaking speed. According to this analysis, it can be assumed that:RS1: The lowest speed was recorded in the case of a cyclist riding individually. In the case of groups of cyclists, a higher speed was observed when riding in line. Only for the largest group consisting of 10 cyclists there was no statistically significant difference between their in-line and two-abreast configuration.RS2: for 1 cyclist riding individually and for a medium group of cyclists riding in line there was a higher mean speed; however, there are no significant differences between the configurations of the cyclist groups in terms of mean overtaking speed on this road.RS3: only the 10 two-abreast configuration presented significant differences among the other cyclist group configurations registering a lower mean speed of the overtaking vehicles. When the groups rode in line, a higher speed was recorded, with the difference between the group of 10 cyclists in line and two abreast being statistically significant.RS4: On this road, there was no difference in mean overtaking speed between 1 cyclist riding individually and a medium-sized group riding in line and two abreast. As for a larger group of 10 cyclists, when riding in line, a higher mean speed of overtaking vehicles was recorded, presenting statistically significant differences with respect to the two-abreast configuration.RS5: the highest overtaking vehicle speed was obtained for a cyclist riding individually, so that when the number of cyclists increases the mean overtaking vehicle speed decreases. A medium-sized group presented a slightly higher overtaking speed when riding in two abreast, whereas for a large group of cyclists there was no difference, taking into account the configuration. The differences between 1 cyclist and 4 two abreast among the other configurations were statistically significant.

In general, wider rural roads presented higher overtaking vehicle speeds, also related with its higher speed limit. When cyclist groups rode in line, a higher overtaking vehicle speed was registered, especially for larger groups.

Additionally, similarly to the case of lateral clearance, a multivariable regression modeling was carried out for each single cyclists’ group configuration (Table 5). As occurs with lateral clearance, the geometric element where the overtaking is performed (tangent, right curve, left curve) is not significant in all cyclists’ group configurations.

The results of the model show that in all cases, the overtaking speed decreases when the passing maneuver is accelerative and when the center line is solid (except in the case of 4 L configuration). Moreover, when the cyclists’ group are larger, there are fewer differences between road segments.

### 3.4. Compliance with Overtaking Standards

The level of non-compliance with minimum overtaking lateral clearance required considering all the overtaking maneuvers performed by PC was 24%. This result is detailed below for each road segment and for each configuration of the group of cyclists.

Figure 8 shows the histograms corresponding to the lateral clearance recorded on each road segment during cyclists overtaking. In this case, all groups of cyclists have been considered together. It can be seen that RS1, with a higher AADT and a stronger speed restriction, presented a higher value of non-compliance with the standards. RS3, which has a red-colored shoulder, had the second highest non-compliance value. RS2 and RS4, despite having the less favorable cross sections, presented lower values of non-compliance with the 1.5 m. Finally, RS5, which has a wider cross section, presented the lowest level of non-compliance.

Regarding the different configurations of groups of cyclists, the percentage of non-compliance with the required 1.5 m when drivers overtake cyclists, considering all road segments together, was 20% for one cyclist riding individually, 22% for 4 and 10 cyclists riding in-line, and 29% for 4 and 10 cyclists riding two abreast.

Figure 9 shows the level of non-compliance considering each group of cyclists on each road segment. On RS1 the level of non-compliance was similar for all cyclist group configurations reaching values above 30%. On RS2 and RS3, the level of non-compliance when the cyclists’ configurations were two abreast was significant higher that otherwise, unlike what happens on RS4. Finally, on RS5, the highest percentage of non-compliance corresponds to the 10 two-abreast configurations.

In most cases, when cyclists rode two abreast, the level of non-compliance was higher.

### 3.5. How Were the Overtaking Maneuvers Performed?

The way in which drivers overtook cyclists was evaluated regarding different characteristics related to the overtaking maneuvers registered. This analysis was performed considering the different road segments and the different configuration of the groups of cyclists. In this way, the influence of these variations on the overtaking maneuver performance was analyzed.

Three characteristics of the overtaking maneuvers were analyzed. The first one is the type of road center line, which can be solid or dashed. The second one is the opposing lane invasion during the overtaking maneuver, and it is related to the risk of frontal crash with oncoming vehicles. Finally, the overtaking strategy was analyzed, that indicates if drivers reduce their speed before overtaking cyclists or not.

Table 6 presents the percentages of these variables per road segment and per group of cyclists. Regarding the road center line, RS1 is not analyzed because it has solid line along its entire length. RS2, with a narrow cross section, presented 41% of overtaking maneuvers performed with solid line. On this road more maneuvers were performed with solid line when one cyclist rode individually, whereas the 10 TA configuration presented the lowest use of solid line. On RS3, with colored shoulder, 34% of the overtaking maneuvers were carried out with solid line and no differences were observed in the use of solid line considering groups of cyclists; only configuration 4 L presented a lower percentage of solid line used during the overtaking maneuver. RS4, which is a narrow road, presented a percentage of use of a solid line of 31%, and presented similar results to RS2, offering a lower number of overtaking maneuvers performed with a solid line when overtaking larger groups of cyclists. Finally, RS5, with a wide cross section, presented the highest percentage of overtaking maneuvers performed with a solid line (60%), and no differences were observed considering the different groups of cyclists.

The opposing lane invasion during the overtaking maneuver presented different values, considering the RS and the group of cyclists. RS1 presented, in general, 69% of maneuvers with opposing lane invasion (partial + total). Especially important is the opposing lane invasion when cyclist groups rode two abreast, as on RS3 and RS5. On RS2, almost all the maneuvers were performed invading the opposing lane. On RS4, almost all the overtaking maneuvers to 4 TA, 10 L, and 10 TA were invading the opposing lane. Therefore, on narrow roads (RS2 and RS4), the opposing lane invasion was present in all overtaking maneuvers.

Concerning the overtaking strategy, on all road segments, the most frequent maneuver is the flying overtaking, presenting lower percentage when overtaking a larger cyclists’ group. RS1 and RS3 presented a high number of piggy backing maneuvers. In the case of RS1, this is probably due to the high AADT. Regarding the accelerative strategy, it was mainly performed on RS2 and RS4, which are narrow roads, and especially to large groups of cyclists riding two abreast.

## 4. Discussion

The results of this paper are based on 1442 overtaking maneuvers by passenger cars registered during a data collection carried out on weekends and weekdays in the morning. Other studies collected data only on weekdays [6,14]. This could be enough for overtaking maneuvers to only one cyclist, but the phenomenon with a cyclists’ peloton is different. Usually, cyclists’ clubs meet for cycling on weekends in the morning and at the same two-lane rural roads. Therefore, the usual drivers are used to meet larger cyclists’ pelotons on Saturdays and Sundays, not on weekdays. Therefore, this research has been carried out on weekends for larger groups and on weekdays for medium groups and individual cyclists. Moreover, the sample could seem small compared with other studies. However, it is important to consider that these studies were based on a cyclist riding alone which is easier to overtake than a peloton of 10 cyclists.

Several studies [6,11,14] equipped the instrumented bicycle with a laser pointer to facilitate the rider to keep a fixed path, and the cyclist should ride within a speed range. In this case, being professionals, the cyclists were free to ride as they would normally do. Moreover, they were not researchers and had few insights on the project. The experiment was thus more naturalistic.

Chosen road segments have different infrastructure and traffic characteristics, thus the analysis among road segments could be extrapolated to the variations due to these characteristics.

RS1 has a very higher AADT and a strong enforcement of the speed limit (60 km/h). These conditions may be the cause for a lower overtaking speed. Moreover, the AADT on the opposing lane is also very high, thus the drivers overtake leaving a small clearance to minimize the risk of a head-on crash. This agrees with Feng et al. [19], who stated that the left-side traffic was associated with significantly less lane-crossing and closer distance to the bike lane or shoulder marking. Under these conditions, riding in smaller groups results in lower overtaking speeds, whereas larger groups riding in line had larger lateral clearances. However, the differences in overtaking speeds are not significant due to the strongly restricted speed limit. On the other hand, RS1 presented the higher value of non-compliance with the 1.5 m, this result can be related to the lower speed limit and higher AADT.

RS2 has a narrow cross-section without a shoulder and a speed limit of 70 km/h. On this road, when cyclist groups rode in line higher clearance was registered than when cyclists rode two abreast, whereas the overtaking speed was practically equal for all cyclists group configurations. This narrow road section generated the invasion of the opposing lane in all overtaking, as well as a higher percentage of accelerative maneuvers.

RS3 and RS4 have the same speed limit (80 km/h) but different geometric characteristics. RS3 has a red shoulder and a favorable cross-section, whereas RS4 has a narrow shoulder. Colored shoulder generates lower lateral clearance than expected regarding the lane and shoulder width, discouraging the use of the colored shoulder. In fact, the colored shoulder presented a higher level of non-compliance with the minimum lateral clearance of 1.5 m.

RS5 is the wider road segment with a higher speed limit. This road presented the higher values of lateral clearance but the higher overtaking vehicle speed. However, RS5 presented the higher percentage of compliance with the 1.5 m in lateral clearance. These results suggest wider roads as a safer for cyclists as stated by other previous studies [6,10,12,19,20]. Previous research also concluded that the distance to the bicycles and the overtaking speed increases when lane width increases [19]. However, the results of this research about the influence of the lane width are not significant because, although there are road segments with different lane widths, it is important to consider the influence of lane and shoulder width together because it is the total space where the drivers and cyclists interact [26].

According to the results, when cyclist groups rode two abreast the percentage of accelerative maneuvers was higher. Accelerative maneuvers were considered safer than flying in previous studies [8,17,21]. The two-abreast configuration of cyclists’ groups presented a lower lateral clearance on most road segments analyzed. These results agree with [23,24], where a group of 3 and 10 cyclists were analyzed, respectively.

Additionally, the level of the non-compliance of Spanish traffic rules was analyzed. Previous research on overtaking a cyclist riding alone on Spanish two-lane rural roads [6] concluded that 36% of motor vehicles overtook under the legal 1.5 m minimum clearance. Regarding the results, the level of non-compliance considering 1 cyclist was lower currently on all roads. This fact is explained by the current high level of driver awareness of cyclists on the road generated by the awareness and education campaigns carried out in recent years in Spain. Considering the overtaking to a group of cyclists, according to the results presented, the level of non-compliance was similar for 1 cyclist than for a group of cyclists riding in line. When cyclists rode two abreast the level of non-compliance was higher due to the higher space the cyclists occupied on the road. In general, the compliance with the 1.5 m depends more on the in line or two-abreast configuration of the group than on the number of cyclists in the group. On the other hand, lane and shoulder width influence the level of compliance with minimum clearance, with both the widest and narrowest roads showing a higher level of compliance, whereas roads of medium width showed the worst results. These medium width roads correspond to RS1 with high AADT and RS3 with a colored shoulder.

Finally, other characteristics on how the overtaking maneuver was performed were analyzed. The center line of the road has a significant effect on the clearance for larger groups of cyclists, and on the overtaking vehicle speed for practically all configurations of groups. On overtaking maneuvers performed with solid line the clearance and the speed were lower. Although a solid center line often means a lower visibility, Spanish regulations allow overtaking cyclists even with solid line [4]. The wider road RS5 is the one with higher use of the solid line to overtake cyclists due to its favorable cross-section. The opposing lane invasion was higher on narrow roads (RS2 and RS4). These results are in line with those of Feng et al. [19], who obtained a decrease in lane-crossing distance when lane width increases.

Based on the results of this study, several lines of further research are proposed. One of them is modeling of the variables that characterize overtaking maneuvers from a safety point of view, which are the lateral clearance and the speed of the overtaking vehicle. These models will need to take into account factors related to the road, the configurations of the groups of cyclists, and the overtaking maneuver itself. On the other hand, it will be interesting to relate the level of risk perceived by cyclists in each overtaking maneuver. These results have already been analyzed for the group of 10 cyclists [24]. However, other sizes and configurations of the group of cyclists should be incorporated.

The results of this study are limited to two-lane rural roads with level terrain and the geometric and traffic conditions analyzed. In fact, the slope of the road has an important effect on the cyclist’s speed. In addition, the results are limited to the cyclists’ group configurations observed in this study. Further research is needed to develop a model to estimate overtaking speed and lateral clearance with these variables and other geometric variables, such as shoulder width and lane width as independent variables.

## 5. Conclusions

The interaction of motor vehicles and bicycles on two-lane rural roads, especially during overtaking maneuvers is one of the main causes of cyclist fatalities. This maneuver is even more challenging when cyclists ride forming a peloton. For this reason, the main objective of the analysis presented in this paper is the characterization of the overtaking maneuver and the study of the differences when the overtaking is performed by cyclists in different group configurations. This characterization was based on the analysis of the variations of the speed developed by the motor vehicles while overtaking and the lateral clearance that they leave between them and the cyclists.

Data collection developed for gathering the necessary data for the analysis was based on semi-professional cyclists riding instrumented bicycles. These cyclists rode along five two-lane rural road segments with different traffic and infrastructure characteristics. At each road segment, they rode in five different group configurations: one individual, four in-line, four two abreast, ten in-line and ten two abreast.

The bicycles were equipped with a small size high-definition video cameras with GPS and a laser device to record the overtaking vehicle speed and the lateral clearance. Due to the small size of the equipment, this data collection methodology could be considered as quasi-naturalistic.

The analysis was based on lateral clearance and overtaking speed as surrogate measures of road safety. Its results show that:The geometric element (tangent, left curve, right curve) where the maneuver is performed has no significant effect on either clearance or overtaking speed.On roads with higher AADT and, therefore, with a higher probability of oncoming vehicle, overtaking maneuver is performed leaving lower lateral clearance. This lateral clearance increases when overtaking larger groups.On road sections without shoulder, motor vehicles cross to the opposing lane to keep a safe lateral clearance and perform the overtaking maneuver at lower speeds when overtake small groups.On road segments with red colored shoulder, motor vehicles overtake leaving lower lateral clearance, especially in two-abreast configurations, and speed is similar to other road segments, except for large groups.Road segments with wider cross-section (shoulder + lane) presents the overtaking maneuvers with higher lateral clearance and higher overtaking speed.Strict speed enforcement results in lower overtaking speed.Narrow roads present higher percentage of accelerative maneuvers, especially when overtaking larger groups.A higher opposing lane invasion was registered on narrow roads and for groups riding in two-abreast configuration.

Additionally, the level of non-compliance with the 1.5 m of lateral clearance of the Spanish traffic rules was analyzed. The level of non-compliance has decreased in recent years. It increases when overtaking a cyclists’ group riding two abreast.

The present study provides an approach to the behavior of drivers when overtake a group of cyclists. The results of the analysis could be used as a basis for the improvement of road segments with high cyclists’ demand. Further research is required to improve the cyclist overtaking characterization, especially on mountainous two-lane rural roads. 

## Figures and Tables

**Figure 1 ijerph-18-12797-f001:**
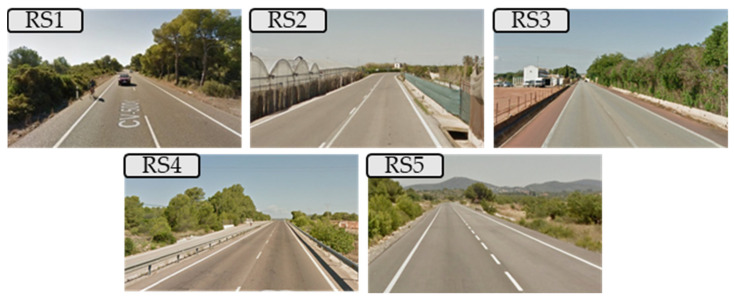
Cross section of the five two-lane rural roads where the study was developed.

**Figure 2 ijerph-18-12797-f002:**
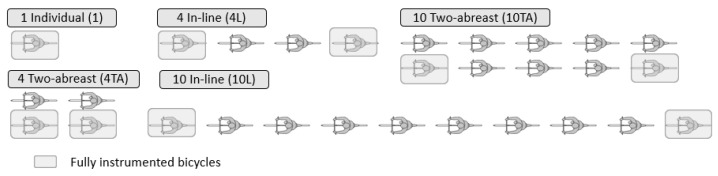
Cyclists group configurations.

**Figure 3 ijerph-18-12797-f003:**
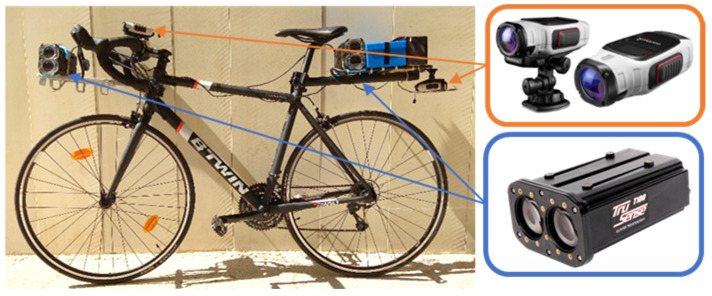
Instrumented bicycle equipped by cameras and a laser device.

**Figure 4 ijerph-18-12797-f004:**
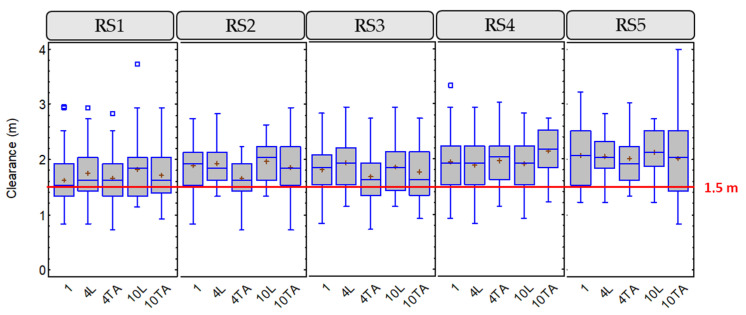
Box-and-whisker plot for lateral clearance, considering road segment and group configuration.

**Figure 5 ijerph-18-12797-f005:**
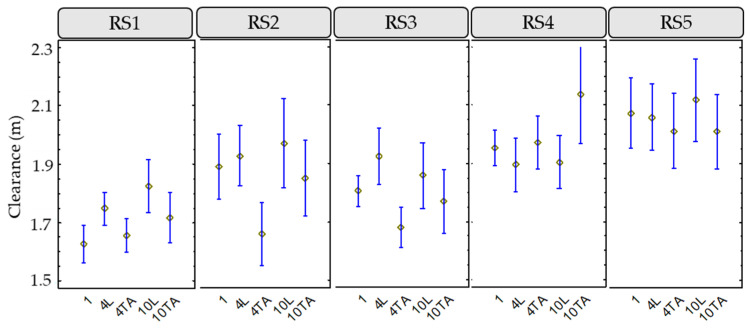
Means and 95% LSD intervals for lateral clearance, considering road segment and peloton configuration.

**Figure 6 ijerph-18-12797-f006:**
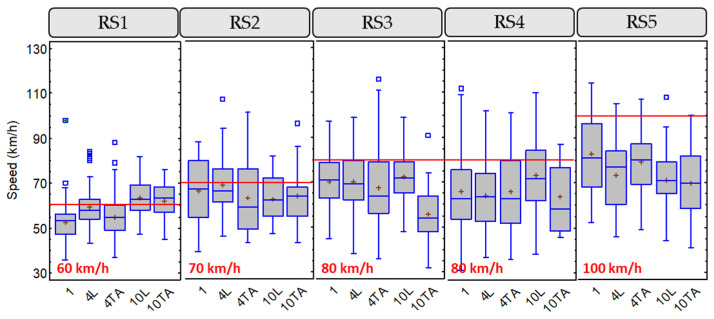
Box-and-whisker plot for overtaking speed, considering road segment and group configuration.

**Figure 7 ijerph-18-12797-f007:**
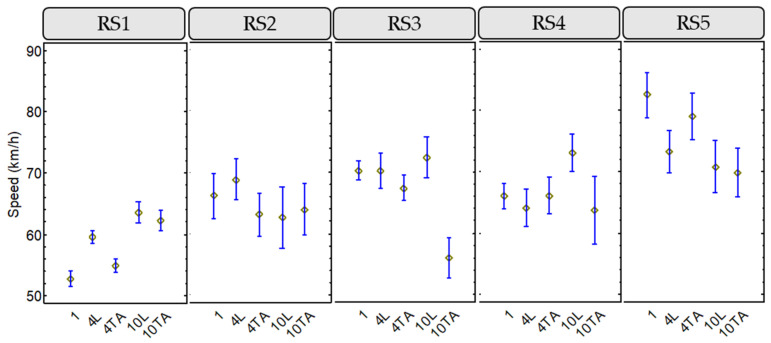
Means and 95% LSD intervals for overtaking speed, considering road segment and peloton configuration.

**Figure 8 ijerph-18-12797-f008:**
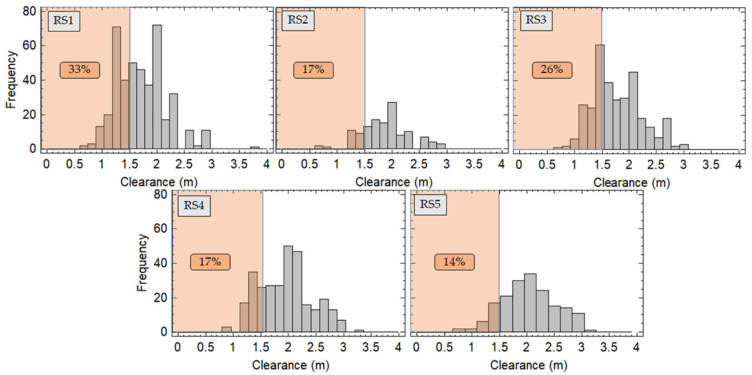
Lateral clearance registered on each road segment and level of non-compliance with Spanish overtaking standards.

**Figure 9 ijerph-18-12797-f009:**
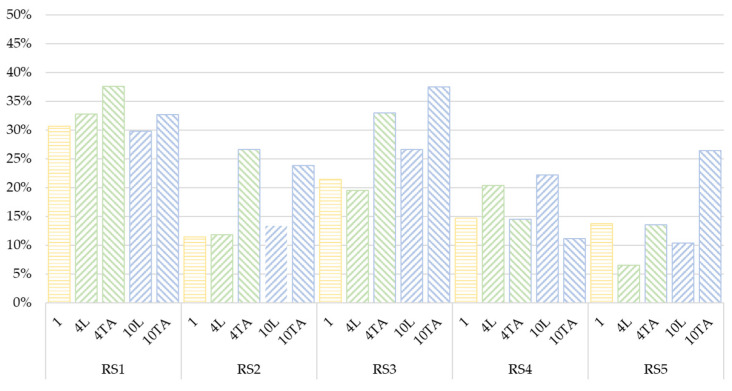
Level of non-compliance of 1.5 m of required lateral clearance per road section and cyclist group.

**Table 1 ijerph-18-12797-t001:** Two-lane rural road characteristics.

Road Section	AADT (veh/Day)	Lane Width (m)	Shoulder Width (m)	Speed Limit (km/h)
RS1	16,310	3.0	1.5	60
RS2	4733	3.5	0.0	70
RS3	5542	3.2	1.5–2.0	80
RS4	6380	3.2	1.0	80
RS5	4797	3.5	1.5	100

**Table 2 ijerph-18-12797-t002:** Number of overtaking maneuvers registered at each road section considering each configuration of the group of cyclists.

	RS1	RS2	RS3	RS4	RS5	Total
1	129	37	160	158	59	543
4 L	131	36	44	59	62	332
4 TA	128	31	88	57	38	342
10 L	51	16	32	63	36	198
10 TA	54	22	36	19	34	165
Total	493	142	360	356	229	1580

**Table 3 ijerph-18-12797-t003:** Number of observations (*N*), mean value, and standard deviation (SD) of lateral clearance and overtaking vehicle speed, considering each overtaking vehicle type.

Vehicle Type	Lateral Clearance			Overtaking Vehicle Speed		
	*N*	Mean (m)	SD (m)	*N*	Mean (km/h)	SD (km/h)
Heavy vehicle (HV)	70	1.73	0.45	70	65.84	14.19
Motorcycle (MC)	56	1.97	0.49	56	72.95	21.21
Passenger car (PC)	1359	1.84	0.48	1350	65.20	14.79
Total	1485	1.84	0.48	1476	65.52	15.11

**Table 4 ijerph-18-12797-t004:** Correlates of lateral clearance in multivariable linear regression modeling.

Independent Variable	Configuration 1			Configuration 4 L			Configuration 4 TA			Configuration 10 L			Configuration 10 TA		
	Coeff.	Std. Error	t Value	Coeff.	Std. Error	t Value	Coeff.	Std. Error	t Value	Coeff.	Std. Error	t Value	Coeff.	Std. Error	t Value
**Intercept**	1.950	0.03	0.33	1.951	0.03	57.62	1987	0.04	43.84	1.927	0.04	47.61	2.166	0.08	26.55
**Road segment**															
RS1	−0.332	0.05	−5.84	−0.204	0.05	−3.86	−0.332	0.06	−5.47	0			0		
RS2	0			0			−0.327	0.09	−3.59	0			0		
RS3	−0.145	0.05	−2.88	0			−0.307	0.06	−4.60	0			−0.218	1.11	−1.92
RS4	0			0			0			0			0		
RS5 (referent)	0			0			0			0			0		
**Centre line**															
Solid line	0			0			0			0			−0.426	0.09	−4.51
Dashed line (referent)	0			0			0			0			0		
**Passing maneuver type**															
Accelerative	0.243	0.11	2.15	0			0			0			0		
Non-accelerative (referent)	0			0			0			0			0		

**Table 5 ijerph-18-12797-t005:** Correlates of overtaking speed in multivariable linear regression modeling.

Independent Variable	Configuration 1			Configuration 4 L			Configuration 4 TA			Configuration 10 L			Configuration 10 TA		
	Coeff.	Std. Error	t Value	Coeff.	Std. Error	t Value	Coeff.	Std. Error	t Value	Coeff.	Std. Error	t Value	Coeff.	Std. Error	t Value
**Intercept**	84.477	2.298	3.704	72.131	1.11	64.42	83.809	2.26	36.97	74.860	1.43	52.2	73.95	1.73	42.71
**Road segment**															
RS1	−29.071	2.598	−11.05	−11.986	1.55	−7.71	−20.093	2.57	−7.79	0			−4.053	2.20	−1.83
RS2	−14.245	3.319	−4.28	0			−13.299	3.08	−4.31	0			0		
RS3	−13.976	2.412	−5.67	0			−13.321	2.51	−5.29	0			−13.591	2.26	−6.00
RS4	−17.982	2.466	−7.19	−6.897	1.96	−3.51	−13.362	2.69	−4.95	0			0		
Rs5 (referent)	0			0			0			0			0		
**Centre line**															
Solid line	−3.197	1.492	−2.20	0			−5.808	1.84	−3.14	−7.701	1.90	−4.04	−4.706	2.08	−2.25
Dashed line (referent)	0			0			0			0			0		
**Passing maneuver type**															
Accelerative	−12.463	3.477	−3.58	−15.602	2.89	−5.38	−18.35	2.00	−9.17	−13.221	2.89	−4.57	−15.388	1.97	−7.80
Non-accelerative (referent)	0			0			0			0			0		

**Table 6 ijerph-18-12797-t006:** Number of overtaking maneuvers recorded (*N*), and percentage of road center line type, opposing lane invasion, and overtaking strategy of overtaking maneuvers performed on each road segment and to each group of cyclists.

			Road Center Line		Opposing Lane Invasion			Overtaking Strategy		
Road	Cyclist Group	*N*	Solid	Dashed	No	Partial	Total	Flying	Accelerative	Piggy Backing
RS1	1	114	100%	0%	54%	44%	2%	82%	3%	16%
	4 L	119	100%	0%	39%	57%	3%	71%	4%	25%
	4 TA	117	100%	0%	9%	78%	14%	68%	17%	15%
	10 L	47	100%	0%	32%	57%	11%	68%	11%	21%
	10 TA	52	100%	0%	8%	63%	29%	56%	19%	25%
RS2	1	35	46%	54%	3%	69%	29%	71%	26%	3%
	4 L	34	41%	59%	0%	74%	26%	82%	12%	6%
	4 TA	30	43%	57%	0%	23%	77%	70%	27%	3%
	10 L	15	40%	60%	0%	40%	60%	60%	33%	7%
	10 TA	21	29%	71%	0%	10%	90%	38%	43%	19%
RS3	1	145	38%	62%	39%	57%	3%	83%	0%	17%
	4 L	41	12%	88%	24%	66%	10%	73%	12%	15%
	4 TA	82	35%	65%	4%	82%	15%	79%	7%	13%
	10 L	30	30%	70%	17%	70%	13%	73%	17%	10%
	10 TA	32	44%	56%	3%	75%	22%	63%	19%	19%
RS4	1	150	32%	68%	21%	65%	14%	95%	3%	2%
	4 L	54	46%	54%	13%	76%	11%	93%	7%	0%
	4 TA	55	24%	76%	2%	56%	42%	80%	16%	4%
	10 L	54	20%	80%	6%	70%	24%	87%	11%	2%
	10 TA	18	22%	78%	0%	17%	83%	50%	50%	0%
RS5	1	51	65%	35%	33%	53%	14%	94%	4%	2%
	4 L	46	59%	41%	39%	57%	4%	98%	0%	2%
	4 TA	37	46%	54%	3%	81%	16%	86%	11%	3%
	10 L	29	69%	31%	31%	69%	0%	76%	3%	21%
	10 TA	34	65%	35%	9%	47%	44%	76%	18%	6%

## Data Availability

Data that support the findings of this study are available from the corresponding author upon reasonable request.

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
