# Peer review of "Driver Behavior When Overtaking Cyclists Riding in Different Group Configurations on Two-Lane Rural Roads"

_ijerph, 2021, doi:10.3390/ijerph182312797_

Round 1

Reviewer 1 Report

This paper studies the driver behavior when overtaking cyclists riding in different group configurations on two-lane rural roads.  Instrumented bicycles were used to collect data on lateral clearance and speed of the overtaking vehicle as well as other variables during overtaking maneuvers. The paper is overall well-written.

In table III, the number of passenger car for lateral clearance and overtaking speed are different (1359 and 1350, resp.). Also, it is mentioned in line 281 that 'the study was performed considering only the 1,442 overtaking maneuvers performed by PC'.
Please explain.

Reviewer 2 Report

This paper mainly analyzes the interaction between vehicles and cyclists on rural two-lane roads. The main purpose of this paper is to analyze the characteristics of vehicle overtaking the cyclists in different group configurations. These characteristics are mainly overtaking speed and lateral clearance. The full text is logical and well-founded, but there are still a few problems:

  1. why the cyclists group configurations range from a single cyclist to group of cyclists of medium size (4 cyclists) and large size (10 cyclists)? Are they representative enough to indicate the law of overtaking?
  2. Under the circumstance of 4 Cyclists and 10 cyclists, there are two bikes equipped with cameras and radars. Are the data measured by these two bikes fused (or corrected)? How is it fused (or corrected)?
  3. The lane widths and shoulder widths of the 5 road sections are not significantly different. Will such similar parameters affect the reliability of the statistical results?
  4. how were the lateral distance (Ld) and relative speed (S) perceived through the camera and radar installed on the bicycle? The acquisition of data is critical to the analysis of statistical results. Please describe in detail.
  5. Does the weather affect the result of overtaking?

Reviewer 3 Report

The paper is very well written, informative and worthy of investigation. I have some minor comments:

  1. The author stated that they did the analysis on weekdays for individual and medium groups while for larger groups they focused on weekends. On weekdays., is there any particular time period the author looked at? If not, then an analysis by the time period might further enhance the contribution of the study.

  1. A policy/elasticity analysis would further increase the impact of the study. The reason is, we can not know the true effect of variables from coefficient itself. Therefore, an elasticity analysis is needed. Doing so, one can easily see which variable is more important specially in overtaking speeds and lateral clearance. You can see but not limited to the following papers if it helps:

Lao, Y., Zhang, G., Wang, Y., & Milton, J. (2014). Generalized nonlinear models for rear-end crash risk analysis. Accident Analysis & Prevention, 62, 9-16.

Bhowmik T., S. Yasmin and N. Eluru (2019), "A Multilevel Generalized Ordered Probit Fractional Split Model For Analyzing Vehicle Speed", Analytic Methods in Accident Research Volume 21, March 2019, Pages 13-31
